# S-phase-independent silencing establishment in *Saccharomyces cerevisiae*

**Davis Goodnight, Jasper Rine\***

Department of Molecular and Cell Biology, University of California, Berkeley, Berkeley, United States

**Abstract** The establishment of silent chromatin, a heterochromatin-like structure at *HML* and *HMR* in *Saccharomyces cerevisiae*, depends on progression through S phase of the cell cycle, but the molecular nature of this requirement has remained elusive despite intensive study. Using high-resolution chromatin immunoprecipitation and single-molecule RNA analysis, we found that silencing establishment proceeded via gradual repression of transcription in individual cells over several cell cycles, and that the cell-cycle-regulated step was downstream of Sir protein recruitment. In contrast to prior results, *HML* and *HMR* had identical cell-cycle requirements for silencing establishment, with no apparent contribution from a tRNA gene adjacent to *HMR*. We identified the cause of the S-phase requirement for silencing establishment: removal of transcription-favoring histone modifications deposited by Dot1, Sas2, and Rtt109. These results revealed that silencing establishment was absolutely dependent on the cell-cycle-regulated interplay between euchromatic and heterochromatic histone modifications.

## Introduction

Inheritance of gene expression state often accompanies the inheritance of genetic content during cell division. Indeed, the eukaryotic replication fork plays host to the enzymes needed to replicate DNA as well as intricate machinery that reassembles chromatin in the wake of replication. However, during development, cell division is also coupled to the rewiring of gene expression patterns that lead to the generation of new cell types. An understanding of chromatin and epigenetics requires an understanding of the mechanisms by which cells can both faithfully transmit chromatin state through cell division, and subvert that inheritance to establish new cell types. The silent chromatin controlling mating-type identity in *Saccharomyces cerevisiae* offers a tractable context for exploring how cell-cycle-regulated chromatin dynamics lead to the establishment of new expression states.

The maintenance of the correct mating type in *Saccharomyces* relies on both the expression of the **a** or α mating-type genes at the *MAT* locus and the heterochromatin-mediated silencing of copies of those same genes at *HML* and *HMR* (*Herskowitz, 1989*). Silencing is dependent on the S̲ilent I̲nformation R̲egulator genes, *SIR1-4*, whose study has led to an understanding of how silencing is achieved (*Gartenberg and Smith, 2016*; *Rine and Herskowitz, 1987*). *HML* and *HMR* are flanked by DNA sequences termed silencers, which recruit the DNA-binding proteins Rap1, Abf1, and ORC. These in turn recruit the Sir proteins via protein-protein interactions. Sir protein recruitment to silencers is followed by the spread of Sir proteins across the multi-kilobase loci by iterative cycles of deacetylation of the tails of histones H3 and H4 by Sir2 and binding of Sir3 and Sir4 to those deacetylated histone tails (*Hecht et al., 1995*; *Hoppe et al., 2002*; *Rusché et al., 2002*).

Despite decades of work, a longstanding puzzle remains at the heart of the mechanism of silencing: cells must pass through S phase to establish silencing, but the identity of the elusive cell-cycle-dependent component is unknown (reviewed in *Young and Kirchmaier, 2012*). Cells with a

*For correspondence:
jrine@berkeley.edu

Competing interests: The authors declare that no competing interests exist.

temperature-sensitive *sir3-8* allele arrested in G1 cannot repress *HMRa1* when switched from the non-permissive temperature to the permissive temperature, but can when allowed to progress through the cell cycle (*Miller and Nasmyth, 1984*). DNA replication per se is not required for silencing establishment. Excised DNA circles bearing *HMR*, but no origin of replication, can be silenced if allowed to pass through S phase (*Kirchmaier and Rine, 2001*; *Li et al., 2001*). Thus, some feature of S-phase, but not DNA replication itself, is crucial for silencing establishment.

Interestingly, low-resolution chromatin immunoprecipitation (ChIP) studies showed that Sir protein recruitment to *HMR* can occur with or without cell-cycle progression, suggesting that Sir protein binding and silencing are not inextricably linked (*Kirchmaier and Rine, 2006*). If Sir proteins can bind to a locus but not silence it, then other molecular changes must be required to create silencing-competent chromatin. In cycling cells undergoing silencing establishment, removal of histone modifications associated with active transcription occurs over several cell cycles (*Katan-Khaykovich and Struhl, 2005*). Furthermore, deletion of genes encoding enzymes that deposit euchromatic histone marks modulates the speed of silencing establishment in cycling cells (*Katan-Khaykovich and Struhl, 2005*; *Osborne et al., 2009*), suggesting that removal of these marks is a key step in building heterochromatin. It is unknown whether the removal of euchromatic marks is related to the S-phase requirement for silencing establishment.

To better understand how chromatin transitions from the active to repressed state are choreographed, we developed an estradiol-regulated Sir3 fusion protein, which, combined with high-resolution ChIP and RNA measurements, allowed precise experimental analysis of silencing establishment with single-cell resolution. We characterized the molecular changes that occur during silencing establishment and identified the genetic drivers of the S-phase requirement for silencing establishment.

## Results

### S phase as a critical window for silencing establishment

Previous studies of silencing establishment have used a variety of strategies to controllably induce silencing, each with its own strengths and weaknesses (see, e.g., *Miller and Nasmyth, 1984*; *Kirchmaier and Rine, 2001*; *Li et al., 2001*; *Lazarus and Holmes, 2011*). We sought a new tool to induce silencing that would allow preservation of the structure of the silencers at *HML* and *HMR* and minimally perturb cell physiology upon induction. To do this, we fused the coding sequence of the estrogen binding domain (*EBD*) of the mammalian estrogen receptor α to *SIR3*, making *SIR3*'s function estradiol-dependent (*Figure 1A*; *Lindstrom and Gottschling, 2009*; *Picard, 1994*). Estradiol addition frees the EBD from sequestration by Hsp90, and hence the induction is rapid because it does not require new transcription or translation (*McIsaac et al., 2011*). *SIR3-EBD* strains grown without estradiol failed to repress *HMR*, mimicking the *sir3Δ* phenotype, while those grown with estradiol repressed *HMR* to a similar degree as wild-type *SIR3* strains (*Figure 1B*).

To test whether the *SIR3-EBD* allele retained the requirement for cell-cycle progression to repress *HMR*, estradiol was added to cells that were either cycling or arrested in G1 by α factor. In cycling cells, silencing establishment of *HMR* occurred gradually over several hours (*Figure 1C*). However, in cells arrested in G1, estradiol led to no measurable repression of *HMR*, even after many hours (*Figure 1D*). Thus, silencing of *HMR* could not occur without progression through the cell cycle, in agreement with prior results using other conditional alleles.

Prior work indicated that S phase is a critical window during which cells may undergo partial silencing establishment (*Kirchmaier and Rine, 2001*; *Lau et al., 2002*; *Miller and Nasmyth, 1984*). Consistent with this, when we arrested cells in G1, then induced *SIR3-EBD* and allowed them to proceed through S phase and re-arrested them at G2/M, *HMR* was repressed ~60% from its starting levels (*Figure 1E*). Crucially, the extent of this partial repression was stable over many hours in these G2/M-arrested cells. Thus, a repression-permissive window or event occurred between G1 and the beginning of mitosis that allowed partial silencing establishment, and further repression was not possible while arrested at G2/M. Indeed, after 3 hr in estradiol, cycling cells were repressed >20 fold from their starting value, compared to only ~3 fold for cells arrested after a single S phase (compare *Figure 1C and E*). This requirement for multiple cell cycles to occur before full gene repression was

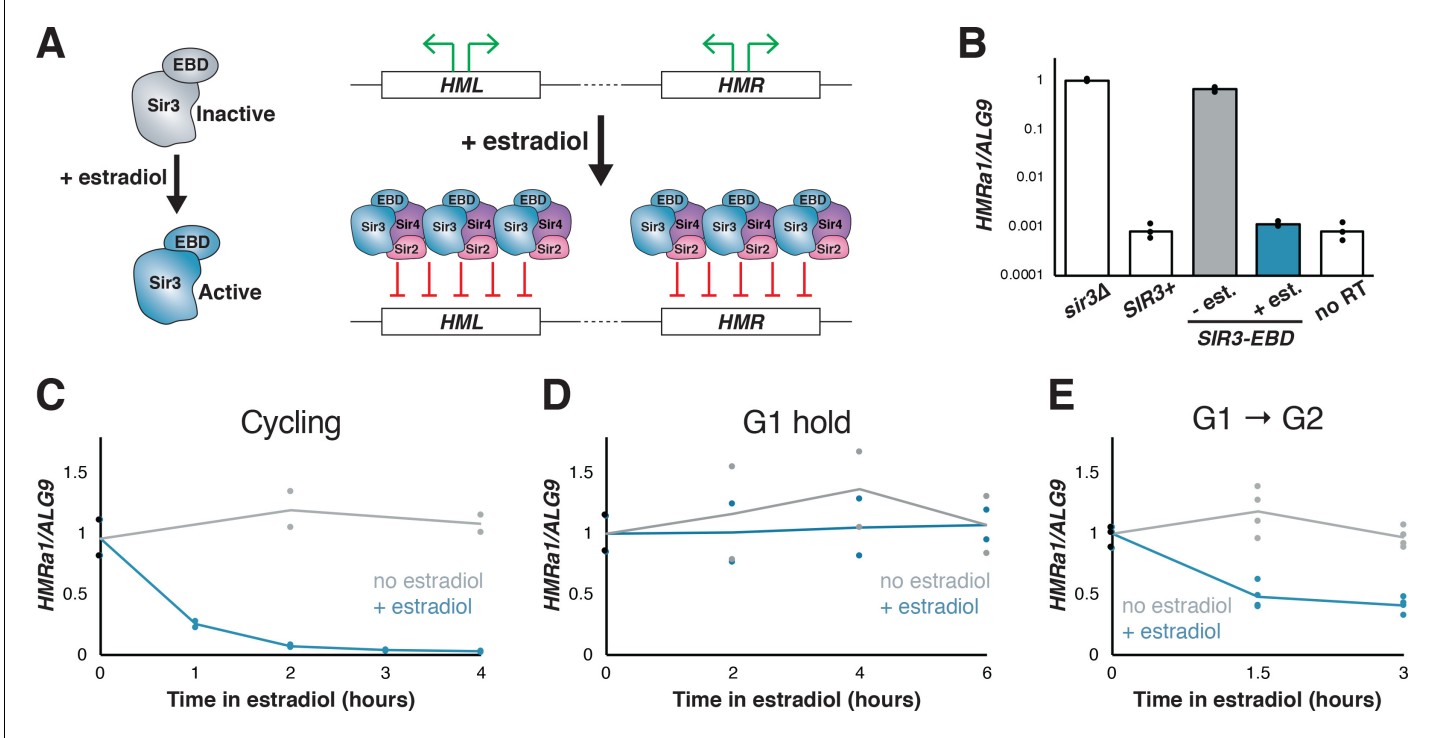

**Figure 1.** Silencing establishment using *SIR3-EBD* required S-phase progression. (**A**) Schematic for *SIR3-EBD* activation. When estradiol is absent, *SIR3-EBD* is kept inactive and *HML* and *HMR* are expressed. Upon addition of estradiol, *SIR-EBD* is activated and *HML* and *HMR* are repressed. (**B**) RT-qPCR of mRNA from *sir3Δ* (JRY12168), *SIR3* (JRY12171), and *SIR3-EBD* (JRY12170) cells grown with ethanol (solvent control) or estradiol (N = 3 for each condition). Also plotted is the non-reverse-transcribed (no RT) value $HMRa1^{no\ RT}/ALG9$ for *SIR3* cells , to demonstrate that *SIR3* cells and *SIR3-EBD* cells grown with estradiol silenced *HMRa1* to essentially the limit of detection. (**C**) *SIR3-EBD* cultures (JRY12169, JRY12170) were grown to mid-log phase, then split and grown in medium with either estradiol or ethanol added. Silencing was monitored by RT-qPCR in a time course after estradiol addition. t = 0 represents the point of estradiol addition for this and subsequent experiments. (**D**) *SIR3-EBD* cultures (JRY12169, JRY12170) were arrested in G1 with α factor, then split, with either ethanol or estradiol added. The arrest was maintained for 6 hr, and silencing was assayed by RT-qPCR throughout. (**E**) *SIR3-EBD* cultures (JRY12169, JRY12170; 2 replicates of each genotype) were arrested in G1 with α factor, then split and released to G2/M by addition of protease and nocodazole in the presence of either ethanol or estradiol. In this and all subsequent figures, dots represent biological replicates, and the bars/lines represent the averages of biological replicates.

The online version of this article includes the following figure supplement(s) for figure 1:

**Figure supplement 1.** Effects of cohesin depletion and *tT(AGU)C* deletion on silencing establishment.

**Figure supplement 2.** Silencing establishment required S phase at *HML*.

consistent with prior studies of silencing establishment both in cell populations and at the single-cell level (*Katan-Khaykovich and Struhl, 2005*; *Osborne et al., 2009*).

Having established the validity of the *SIR3-EBD* fusion as a tool for studying silencing, we revisited two mutants that have been reported to bypass cell-cycle requirements for silencing establishment. In one study, depletion of the cohesin subunit Mcd1/Scc1 allowed for increased silencing in G2/M-stalled cells (*Lau et al., 2002*). In another study, deletion of a tRNA gene adjacent to *HMR*, termed *tT(AGU)C*, which is known to bind cohesin, was found to allow partial silencing establishment at *HMR* without S-phase progression (*Lazarus and Holmes, 2011*). Using *SIR3-EBD* in combination with an auxin-inducible degron (AID)-tagged Mcd1, we found no effect of depleting cohesin or deleting the tRNA gene in regulating silencing establishment (*Figure 1—figure supplement 1*). Thus, at least for strains using *SIR3-EBD*, the genetic basis for the cell-cycle requirement for silencing establishment at *HMR* remained to be determined. Possible explanations for the discrepancies between our results and earlier reports are discussed below.

Our finding that *tT(AGU)C* did not regulate silencing establishment led us to reconsider the broader claim that *HMR* is distinct from *HML* in its requirement of S phase passage for silencing

establishment (*Lazarus and Holmes, 2011*; *Ren et al., 2010*). Earlier silencing establishment assays at *HML* were complicated by the strong silencing-independent repression of *HMLα1* and *HMLα2* by the **a**1/α2 repressor (*Herskowitz, 1989*; *Siliciano and Tatchell, 1986*): when both **a** and α information are expressed in the same cell, the **a**1 and α2 proteins form a transcriptional repressor whose targets include the *HMLα* promoter. Unless strains are carefully designed, conventional measures of silencing will also inadvertently measure this silencing-independent repression. To circumvent this limitation, we constructed an allele of *HML* with nonsense mutations in both *α1* and *α2*, so that the α1 and α2 proteins were never made, even when *HML* was de-repressed. This modification also allowed us to use α factor to arrest cells while studying silencing establishment at *HML*, which was not possible before because expression of either the α1 or α2 protein renders cells insensitive to α factor. We also introduced additional single nucleotide polymorphisms into the regions of *HML* that are homologous to *HMR*, to allow unambiguous assignment of high-throughput sequencing reads to the two loci (see below). We refer to the mutant locus as *HML\** and the mutant alleles as *hmlα1\** and hmlα2\* hereafter.

When cells with *HML\** and *SIR3-EBD*, were arrested in G1 and then treated with estradiol, they were unable to silence *hmlα1\** or *hmlα2\** while kept in G1 (*Figure 1—figure supplement 2A and B*). Interestingly, expression of *hmlα1\** and hmlα2\* increased markedly over the course of the α-factor arrest. This α-factor-dependent hyper-activation was observed even in *sir3Δ* cells in which no silencing occurs (*Figure 1—figure supplement 2E and F*). We identified two previously-unreported binding sites for Ste12, the transcription factor activated by mating pheromone, in the bidirectional *α1/α2* promoter, which explains the increased expression when cells are exposed to α factor (*Figure 1—figure supplement 2G*; *Dolan et al., 1989*). Both *hmlα1\** and *hmlα2\** decreased in expression following release from G1 to G2/M (*Figure 1—figure supplement 2C and D*), suggesting that S phase was required for partial silencing establishment at *HML*. Notably, the fold change in expression that followed a single passage through S phase was not identical among *HMRa1*, *hmlα1\**, and *hmlα2\**. Thus, some S-phase-dependent process was important for silencing both *HML* and *HMR*, but the effects of that process varied in magnitude between these two loci.

## Silencing establishment occurred through a partially repressive intermediate

We were interested in the partial silencing observed in cells that transited through a single S phase after *SIR3-EBD* induction, in which transcription of *HMRa1* was down ~60% (see *Figure 1E*). This appearance of a stable intermediate level of silencing could reflect either of two distinct phenomena at the single-cell level (*Figure 2A*). One possibility was that cells have a ~60% chance of establishing stable heterochromatin during the first S phase after *SIR3-EBD* induction and a ~40% chance of failing to do so. This possibility would resonate with the behavior of certain mutants, for example *sir1Δ*, wherein silent loci can exist in one of two epigenetic states: stably repressed or stably de-repressed, with rare transitions between the two (*Pillus and Rine, 1989*; *Xu et al., 2006*). Alternatively, every cell might reach a partially repressive chromatin state at *HMR* during the first S phase during silencing establishment.

To distinguish between these possibilities, we used single-molecule RNA fluorescence in-situ hybridization (smRNA-FISH) to quantify the expression of *HMRa1* during the establishment process. If silencing establishment proceeded via individual cells transitioning between the discrete 'ON' and 'OFF' states during S phase, we would expect an accumulation of cells with zero transcripts during the establishment of silencing, with no change in the average number of transcripts in those cells still expressing *HMRa1*. However, if silencing establishment proceeded via partially repressive intermediates in individual cells, we would expect a shift downward in the mean number of transcripts per cell (*Figure 2A*). In both cases, cells with zero transcripts would accumulate over time.

As expected, *SIR3-EBD* cells arrested in G1 without estradiol had similar numbers of *HMRa1* transcripts as *sir3Δ* cells (*Figure 2—figure supplement 1A*). When we added estradiol and allowed the cells to go through S phase to G2/M, the decrease in transcript number in the population of cells analyzed closely mirrored the results we obtained using RT-qPCR, confirming that our single-molecule analysis was consistent with bulk measurements (*Figure 2—figure supplement 1B*, compare to *Figure 1E*). This decrease occurred via a reduction in the average number of transcripts per cell, and not simply by an increase in the number of cells with zero transcripts (*Figure 2B*, *Figure 2—figure*

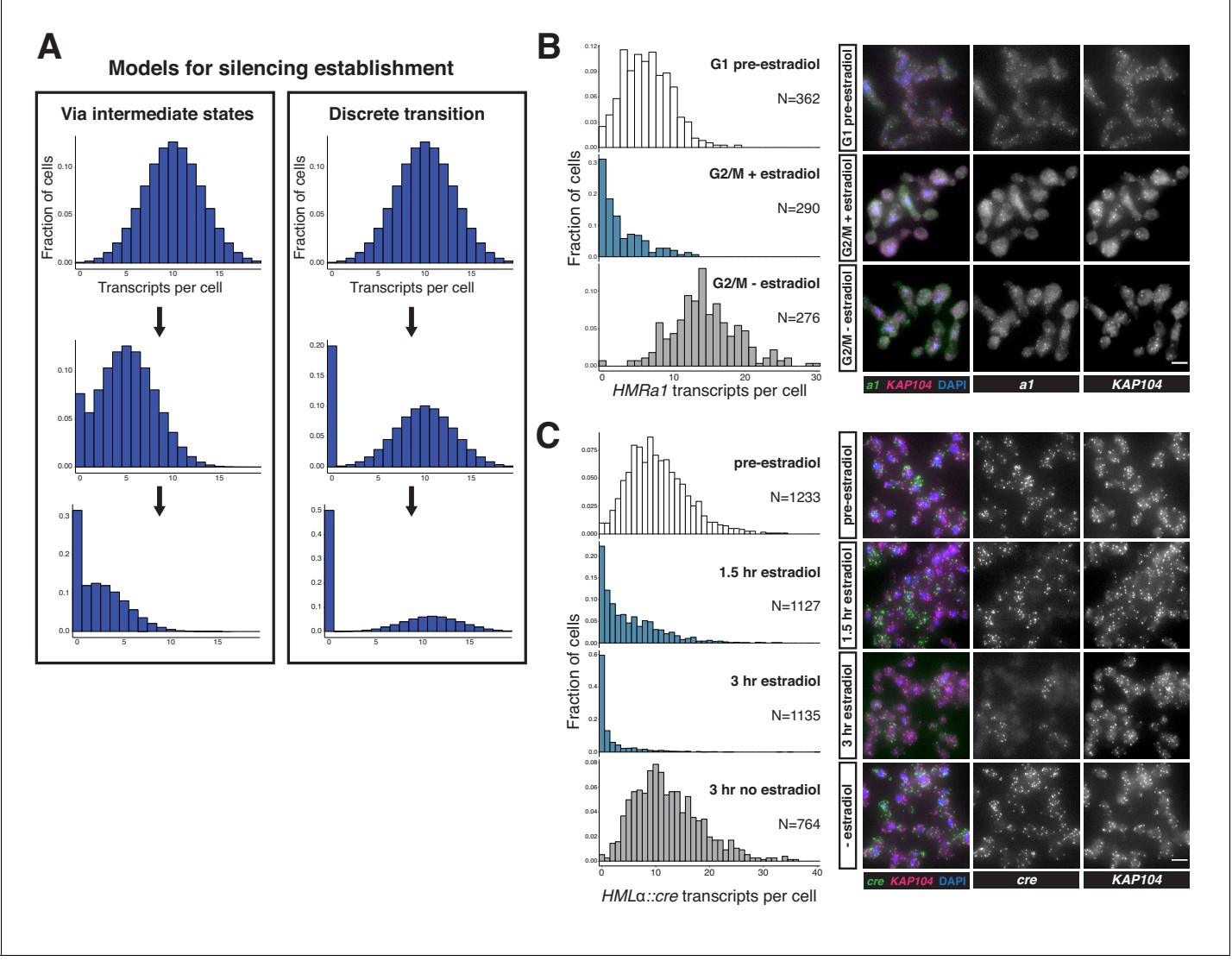

**Figure 2.** Silencing establishment proceeded via gradual repression in individual cells. (A) Potential models for silencing establishment. Before silencing establishment (top), mRNA transcripts are present as a distribution around a mean. If silencing establishment occurred via intermediate states (left), the mean number of transcripts per cell would decrease over time, with complete silencing, that is zero transcripts per cell, occurring as the probability distribution shifted toward the y axis. If silencing establishment occurred via discrete transitions (right), an increasing fraction of cells would have zero transcripts over time, but the distribution of cells with >0 transcripts would retain the same shape. (B) smRNA-FISH for *HMRa1* during silencing establishment after 1 s phase. A *SIR3-EBD* culture (JRY11762) was arrested in G1 with α factor ('G1 pre-estradiol'), then split and released to G2/M by addition of protease and nocodazole in the presence of either estradiol or ethanol. Samples were collected 2 hr after estradiol addition. (C) smRNA-FISH for *HMLα::cre* during silencing establishment in cycling cells. A *SIR3-EBD* strain bearing the *HMLα::cre* reporter (JRY12514) was grown to mid-log phase ('pre-estradiol'), then the culture was split in two, with one sub-culture receiving estradiol and the other receiving ethanol. Samples were collected for smRNA-FISH at t = 1.5 hr and t = 3 hr after estradiol addition. For both (B) and (C), the images displayed are representative maximum-intensity Z-projections. The data shown in (B) and (C) each represent one of two replicate experiments, for which the other replicate is shown in *Figure 2—figure supplements 1* and *2*, respectively. Scale bars = 5 μm.

The online version of this article includes the following figure supplement(s) for figure 2:

**Figure supplement 1.** Gradual silencing establishment at *HMR*.

**Figure supplement 2.** Gradual silencing establishment at *HMLα::cre.*

*supplement 1C and D*). Thus, individual cells undergoing silencing establishment at *HMR* formed partially repressive heterochromatin after a single S phase.

To determine whether the stepwise repression seen at *HMR* was a general feature of silencing establishment or if it was particular to *HMR* and/or the cell-cycle conditions tested, we performed an analogous experiment using the *HMLα::cre* allele that has been previously characterized by smRNA-FISH (*Dodson and Rine, 2015*). In this experiment, we analyzed the number of *cre* transcripts per cell over time following induction of *SIR3-EBD*, without any cell-cycle perturbations. Silencing establishment at *HML* also occurred via partially repressive intermediate states (*Figure 2C*, *Figure 2—figure supplement 2A and B*). The degree of repression observed by smRNA-FISH was quantitatively similar to the measurement of the same gene by RT-qPCR (*Figure 2—figure supplement 2C and D*). Together, these results suggested that silencing establishment proceeded via the gradual repression of genes by the Sir proteins, and that there were specific windows of the cell cycle during which transcriptional tune-down could occur.

## Extensive Sir protein binding could occur without gene repression

The gradual silencing establishment described above might be achieved via increased Sir protein recruitment to *HML* and *HMR* during each passage through a specific cell-cycle window. Alternatively, Sir protein recruitment might be independent of the cell cycle, in which case passage through the cell cycle would favor repression via a step occurring after Sir protein recruitment. To test whether Sir protein recruitment was limited in the cell cycle, we performed ChIP-seq on myc-tagged Sir4 during silencing establishment. The tagged Sir4-myc is functional for silencing (*Figure 1B*), and its localization at *HML* and *HMR* is indistinguishable from Sir2-myc and Sir3-myc in wild-type cells (*Thurtle and Rine, 2014*).

We developed a protocol for ChIP-seq using MNase-digested chromatin, which resulted in increased signal-to-noise over standard sonication-based ChIP-seq (*Figure 3*, *Figure 3—figure supplements 1* and *2*). One limitation of this approach is that MNase can digest non-nucleosomal DNA, and thus the silencers at *HML** and *HMR* and the tRNA gene adjacent to *HMR*, which are not nucleosome-bound, are under-recovered relative to sonication ChIP (*Figure 3—figure supplement 2*).

As expected, ChIP-seq on cells with wild-type *SIR3* revealed strong binding of Sir4-myc throughout *HML** and *HMR* (*Figure 3A and B*). *SIR3-EBD* cells grown with estradiol gave profiles that were indistinguishable in both the strength of binding and the location of binding. In *sir3Δ* cells and *SIR3-EBD* cells grown without estradiol, some Sir4-myc binding across *HML** and *HMR,* though severely reduced, was still evident (*Figure 3A and B*). This weak binding was observed in multiple replicates with different crosslinking times and was not observed in cells with untagged Sir4 (*Figure 3—figure supplement 1A*). We also performed ChIP-seq for Sir3-EBD using an antibody to the estrogen receptor, and found that its binding was strongly dependent on the presence of estradiol, and its binding pattern was indistinguishable from Sir4-myc (*Figure 3—figure supplement 3*). The apparent weak Sir3-EBD signal at *HML** and *HMR* in the absence of estradiol did not drive the weak Sir4-myc binding described above, as the same Sir4-myc binding was observed in *sir3Δ* cells as in *SIR3-EBD* cells grown without estradiol.

ChIP-seq data from *SIR3-EBD* cells arrested in G1 without estradiol revealed the same weak enrichment of Sir4-Myc at *HML** and *HMR* that we observed in cycling cells (*Figure 3C*, *Figure 3—figure supplement 1C*). However, upon addition of estradiol in cells kept in G1, we saw a strong increase in Sir4-myc binding across the loci (*Figure 3C*, *Figure 3—figure supplement 1C*). The increase in Sir4-myc binding to *HMR* was not associated with any change in expression of *HMRa1*, which remained completely de-repressed (*Figure 3—figure supplement 1B*). Hence, Sir protein binding across *HML** and *HMR* was not sufficient to lead to gene silencing. When cells were allowed to pass from G1 to G2/M, the resulting partial silencing was correlated with an increase in Sir4-myc binding at *HML** and *HMR* (*Figure 3C*, *Figure 3—figure supplement 1C*). Thus, Sir proteins binding throughout *HML** and *HMR* in absence of cell-cycle progression achieved no repression, and some S-phase-dependent process promoted further binding and partial repression. Together, these data revealed the existence of cell-cycle-regulated steps beyond Sir binding required to bring about silencing.

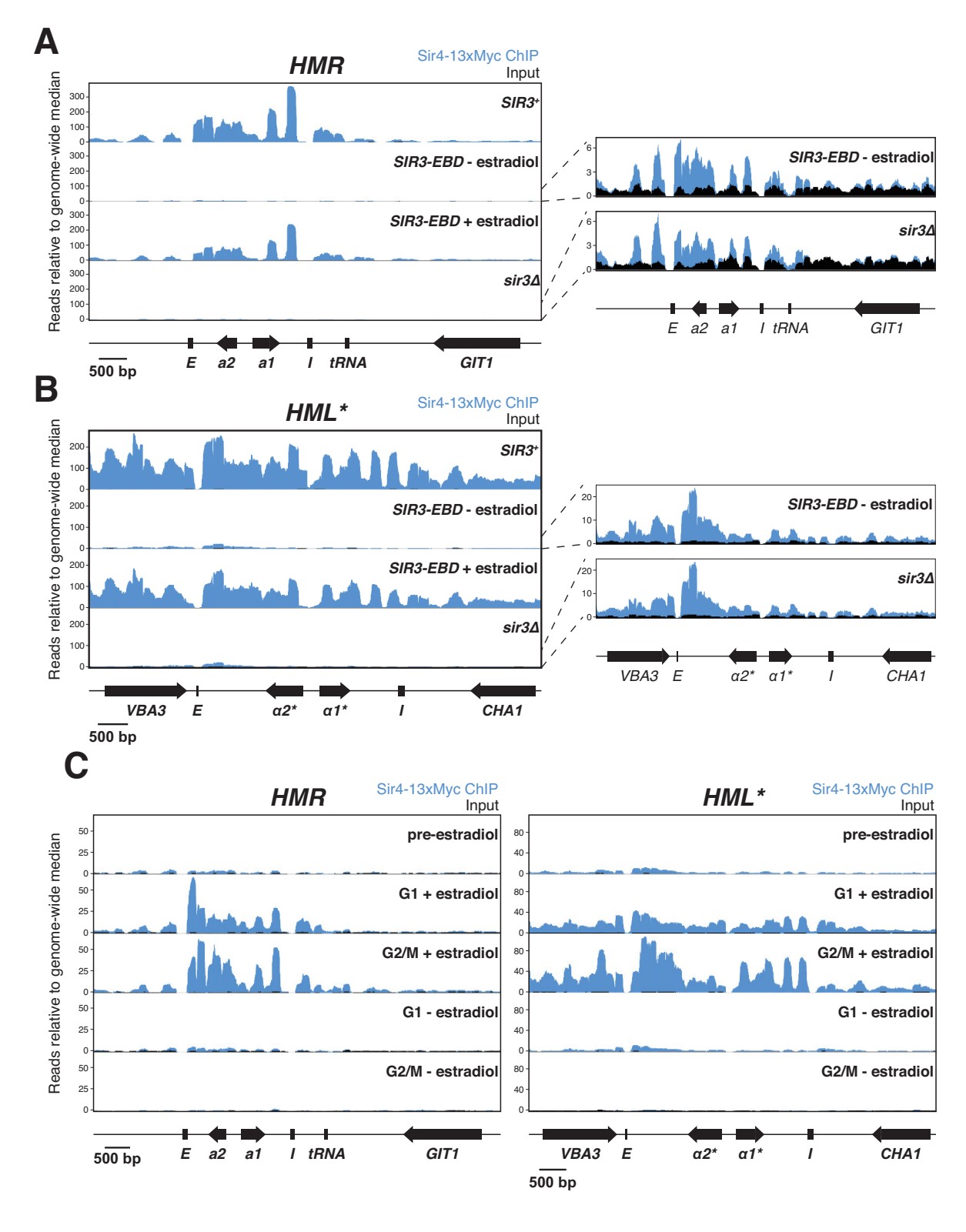

**Figure 3.** Sir protein binding and silencing were separable phenomena. All panels show Sir4-13xMyc ChIP-seq signal in blue and input in black. Read counts were normalized to the non-heterochromatin genome-wide median. IP and input values are plotted on the same scale. (**A**) Left, ChIP-seq for Sir4-13xMyc at *HMR* in strains with *SIR3* (JRY12172), *sir3Δ* (JRY12168), and *SIR3-EBD* (JRY12170) grown with or without estradiol and fixed for 60 min in formaldehyde. Right, same data as the left panel for *sir3Δ* and *SIR3-EBD* without estradiol, enlarged to show IP levels above input. (**B**) Same as (**A**), but

*Figure 3 continued on next page*

*Figure 3 continued*

showing data from *HML*. (**C**) ChIP-seq for Sir4-13xMyc during silencing establishment at *HMR* (left) and *HML** (right). Cultures of *SIR3-EBD* cells (JRY12169) were arrested in G1 with α factor ('pre-estradiol'), then split four ways. Two sub-cultures were maintained in G1 in medium with estradiol or ethanol ('G1 + estradiol' and 'G1 - estradiol'). The other two sub-cultures were released to G2/M by addition of protease and nocodazole; and received either estradiol or ethanol ('G2/M + estradiol' and 'G2/M - estradiol'). After 3 hr in medium with estradiol or ethanol, cultures were fixed in formaldehyde for 15 min and collected for ChIP-seq. Data shown represent one of two replicates, with the other shown in *Figure 3—figure supplement 1C*.

The online version of this article includes the following figure supplement(s) for figure 3:

**Figure supplement 1.** Silencing establishment ChIP-seq.
**Figure supplement 2.** ChIP-seq with sonicated chromatin.
**Figure supplement 3.** ChIP-seq for Sir3-EBD.

## Removal of H3K79 methylation was a critical cell-cycle-regulated step in silencing establishment

Given that silencing at *HML* and *HMR* was established only during a discrete window of the cell cycle, the key issue was to identify what molecular event(s) occurred during this window and why it/they were limited in the cell cycle. A mutant that could establish silencing while arrested in G1 would potentially identify that molecular event.

The histone methyltransferase Dot1 has several characteristics that suggest it might act as an antagonist of silencing establishment. Dot1 methylates histone H3 on lysine 79 (H3K79), which interferes with Sir3 binding to nucleosomes (*Altaf et al., 2007*; *Armache et al., 2011*; *van Leeuwen et al., 2002*; *Yang et al., 2008*). Dot1 is unique among yeast histone methyltransferases in lacking a counteracting demethylase that removes H3K79 methylation. Thus, removal of H3K79 methylation can be achieved only through turnover of the histones that bear it, such as occurs during S phase, when new histones are incorporated that lack H3K79 methylation (*De Vos et al., 2011*). Indeed, *dot1Δ SIR3-EBD* cells arrested in G1 robustly repressed *HMRa1*, *hmlα1**, and *hmlα2** upon addition of estradiol (*Figure 4A*, *Figure 4—figure supplement 1A and B*). This phenotype was not limited to *SIR3-EBD* strains. Strains bearing the temperature-sensitive *sir3-8* allele and *dot1Δ* could also establish silencing in G1 when shifted from the non-permissive temperature to the permissive temperature (*Figure 4E*). Thus, removal of H3K79me from *HML* and *HMR* was one crucial S-phase-specific step during silencing establishment.

To test whether the *dot1Δ* phenotype was due specifically to methylation at H3K79, both copies of histone H3 were mutated to encode arginine at position 79 (*H3K79R*), a mimic for the non-methylated state. This mutant also allowed for robust silencing establishment in G1, in fact, to a stronger degree than *dot1Δ* (*Figure 4C*). Strains with H3K79 mutated to leucine (*H3K79L*) or methionine (*H3K79M*) failed to establish silencing even after passage through S phase (*Figure 4—figure supplement 1C and D*), confirming the importance of the positive charge on H3K79 in silencing. Notably, even though G1-arrested *H3K79R* cells could strongly repress *HMRa1* (~15 fold), this was still incomplete relative to fully silenced cells, which repressed *HMRa1* >1000 fold (see *Figure 1B*). Thus, either increased time or cell cycle progression promoted silencing establishment even in absence of H3K79 methylation.

In addition to promoting S-phase-independent silencing establishment, *dot1Δ* and *H3K79R* cells that passed from G1 to G2/M also repressed *HMRa1* more robustly than did wild-type cells transiting the same cell-cycle window (*Figure 4B* and *Figure 4D*, compare to *Figure 1E*). S-phase passage markedly increased the speed of silencing establishment in *dot1Δ* cells, though the ultimate degree of repression was similar whether cells passed through S phase or stayed in G1 (compare *Figure 4A and B*). Thus, some feature of S phase still promoted silencing establishment in cells lacking H3K79 methylation.

To understand how H3K79 methylation prevented silencing establishment, we performed ChIP-seq for Sir4-myc in *dot1Δ* cells undergoing silencing establishment in the absence of cell-cycle progression. G1-arrested *dot1Δ* cells already displayed clear partial silencing establishment after 1.5 hr in estradiol (*Figure 4—figure supplement 1E*). However, the level of Sir4-myc recruitment to *HML** and *HMR* at this early time point was similar to the recruitment observed in wild-type cells, in which

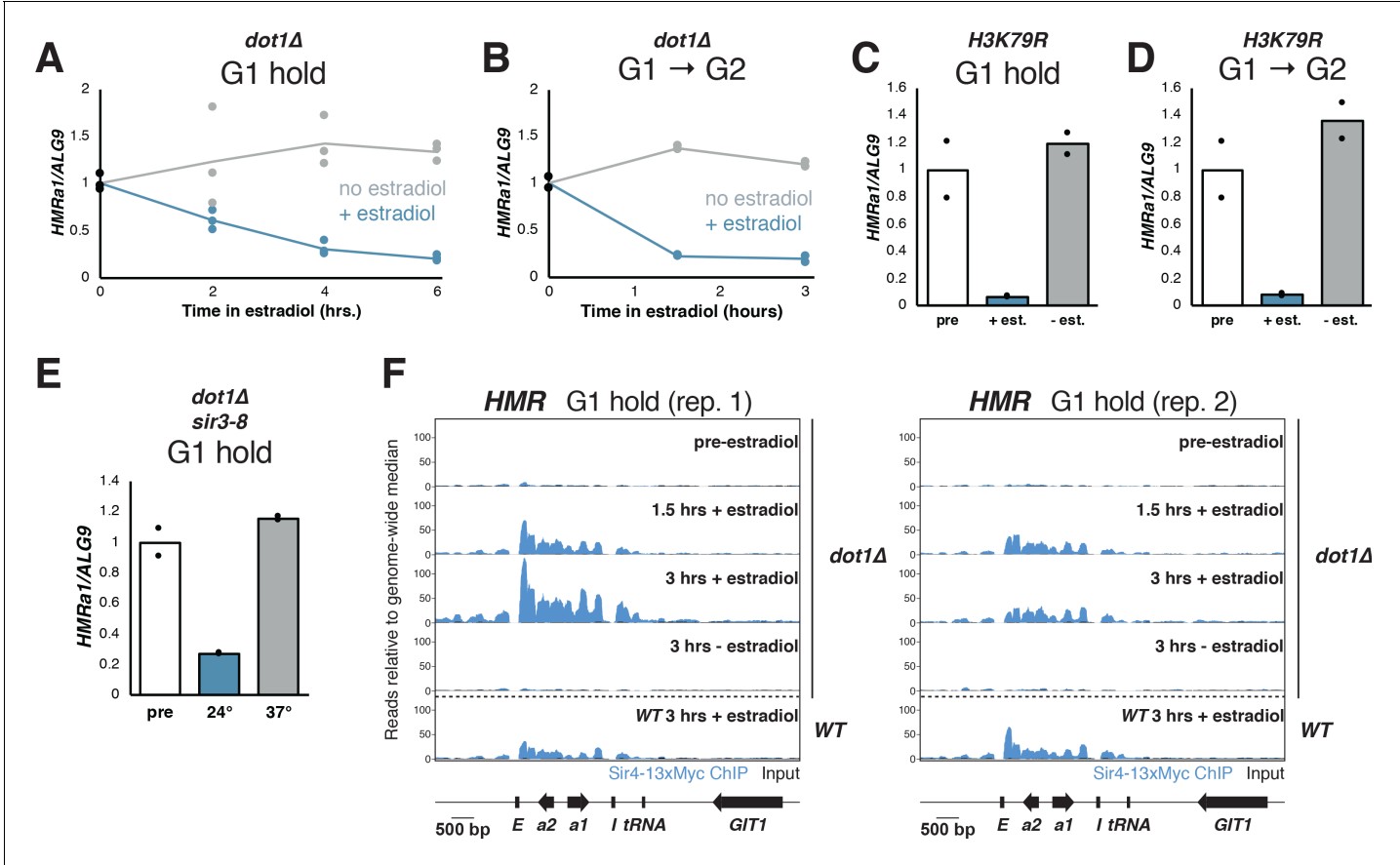

**Figure 4.** Cells without H3K79 methylation established silencing without cell-cycle progression. (A) Cultures of *dot1Δ* cells (JRY12443, JRY12445) were arrested in G1 with α factor, then split, with half receiving ethanol and the other half receiving estradiol. Silencing was monitored by RT-qPCR over time after estradiol addition. (B) *dot1Δ* mutants were arrested in G1 with α factor, then released to G2/M by addition of protease and nocodazole, and either ethanol or estradiol. Silencing was monitored by RT-qPCR over time after estradiol addition. (C) Cultures of cells in which lysine 79 was mutated to arginine in both *HHT1* and *HHT2* in two isogenic strains (*H3K79R*; JRY12851, JRY12852) were arrested in G1 with α factor, then split, with one sub-culture receiving ethanol and the other receiving estradiol. Silencing was assayed by RT-qPCR after 6 hr in ethanol. (D) *H3K79R* cells were arrested in G1 with α factor, then released to nocodazole with protease and nocodazole and either estradiol or ethanol. Silencing was assayed by RT-qPCR after 3 hr in estradiol. The pre-estradiol sample for this experiment was the same culture used in (C). (E) Cultures of *dot1Δ sir3-8* cells (JRY12859, JRY12890) were grown at the non-permissive temperature for *sir3-8* (37°C) and arrested in G1 with α factor, then split, with half shifted to the permissive temperature (24°C) and other half staying at the non-permissive temperature. Silencing was assayed by RT-qPCR after 6 hr. (F) Cultures of *dot1Δ* cells (JRY12443, JRY12444) were arrested in G1 with α factor ('pre-estradiol'), then split, with half the culture receiving ethanol, and the other half receiving estradiol. After 1.5 hr and after 3 hr, samples were fixed for 15 min in formaldehyde and collected for ChIP. Sir4-13xMyc ChIP-seq signal is in blue and input in black, each normalized to the non-heterochromatin genome-wide median and plotted on the same scale. Also displayed are two replicates of wild-type G1 cells after 3 hr in estradiol from *Figure 3C* and *Figure 3—figure supplement 1C*.

The online version of this article includes the following figure supplement(s) for figure 4:

**Figure supplement 1.** Silencing establishment in *dot1Δ* cells.

**Figure supplement 2.** H3K79 trimethylation dynamics during silencing establishment.

no gene repression had occurred after 3 hr (*Figure 4F*, *Figure 4—figure supplement 1F*). There-fore, even though *dot1Δ* and wild-type cells had indistinguishable levels of Sir binding, that binding gave rise to different transcriptional effects. Thus, removal of H3K79me did not regulate silencing establishment by controlling Sir protein binding.

We next tested explicitly whether H3K79 methylation depletion can occur during a G1 arrest and during passage through S phase. As noted previously, in wild-type cells, H3K79 trimethyla-tion (H3K79me3) is almost completely absent from *HML** and *HMR*, but in *sir3Δ* mutants,

H3K79me3 is present at both loci (*Figure 4—figure supplement 2A and B*). When *SIR3-EBD* was induced in cells arrested in G1, no change in H3K79me3 was observed (*Figure 4—figure supplement 2*), even though robust Sir protein recruitment could occur (*Figure 3C*). Following S phase, though, H3K79me3 was partially depleted from *HMR* and from *hmlα1\**, but not from *hmlα2\** (*Figure 4—figure supplement 2*). Thus, Sir protein binding in G1 was insufficient to change H3K79me3 levels at *HML* and *HMR*. The first depletion of this mark occurred concomitantly with S phase.

### *SAS2* and *RTT109* contributed to limiting silencing establishment to S phase

The crucial role of H3K79 methylation removal in silencing establishment led us to consider other chromatin modifications that might regulate silencing establishment. Two histone acetyltransferases, Sas2 and Rtt109, were especially interesting given the S-phase dynamics of the marks they deposit and their known relevance to silencing. Sas2, the catalytic component of the SAS-I complex, acetylates H4K16 during S phase (*Kimura et al., 2002*; *Meijsing and Ehrenhofer-Murray, 2001*; *Reiter et al., 2015*; *Suka et al., 2002*). The removal of H4K16 acetylation by Sir2 is the central histone modification associated with silencing (*Imai et al., 2000*; *Johnson et al., 1990*; *Landry et al., 2000*; *Park and Szostak, 1990*). Rtt109 acetylates newly-incorporated histone H3 at lysines 9 and 56 during S phase, and this acetylation is largely removed by Hst3 and Hst4 by the time of mitosis (*Adkins et al., 2007*; *Celic et al., 2006*; *Driscoll et al., 2007*; *Fillingham et al., 2008*; *Schneider et al., 2006*). Mutations in *SAS2* and *RTT109* have both been shown to have subtle silencing phenotypes (*Imai et al., 2000*; *Miller et al., 2008*).

Interestingly, both *sas2Δ* and *rtt109Δ* mutations led to partial repression of *HMR* upon *SIR3-EBD* induction in cells arrested in G1 (*Figure 5A*). The magnitude of this effect was weaker than in *dot1Δ* cells but highly significant. Cells without *RTT109* grew slowly and were less sensitive to α factor than wild-type cells, so we cannot exclude the possibility that a population of *rtt109Δ* cells passed through S phase during the experiment, contributing to the observed phenotype (*Figure 5—figure supplement 1*). When combined with *dot1Δ*, both *sas2Δ* and *rtt109Δ* led to a further increase in silencing establishment. Thus, *SAS2* and *RTT109* impeded silencing establishment by a different mechanism than *DOT1*. Silencing establishment in cells lacking both *SAS2* and *RTT109* was not significantly different from that of the single mutants. Strikingly, triple mutant *sas2Δ rtt109Δ dot1Δ* strains established silencing no better than single-mutant *dot1Δ* cells. Interestingly, the G1 phenotypes we observed at *HMR* for *dot1Δ*, *sas2Δ*, and *rtt109Δ* single mutant cells were largely similar at *hmlα1\** (*Figure 5B*). Altogether, these findings demonstrate that *SAS2*, *RTT109*, and *DOT1* inhibit silencing establishment outside of S phase.

## Discussion

In this study, we resolved why silencing establishment requires cell cycle progression. These results highlighted the value of studying the dynamics of silencing both in populations of cells and at the single cell level. By monitoring changes in chromatin and changes in expression simultaneously, we documented effects that were elusive at steady state, but critical for a mechanistic understanding of the process. We found that the cell-cycle-dependent removal of euchromatic marks was a major driver of a cell's ability to establish stable heterochromatin. Interpretation of our results required critical reassessment of some earlier results.

### Silencing establishment occurred by tuning down transcription in individual cells after Sir proteins were bound

The classic model for silencing establishment involves two steps: nucleation of Sir proteins at the silencers, followed by spreading of Sir proteins from silencers via the stepwise deacetylation of nucleosomes by Sir2 and subsequent binding of Sir3 and Sir4 to deacetylated positions of H3 and H4 tails (*Hecht et al., 1995*; *Hoppe et al., 2002*; *Rusche et al., 2003*; *Rusché et al., 2002*). In the classic model, individual Sir proteins are recruited to the silencers, but the spread across the locus is dependent on all three proteins Sir2/3/4, with both continuous Sir protein binding and histone deacetylation being required for gene repression (*Johnson et al., 2009*; *Yang and Kirchmaier, 2006*). The binding of a Sir2/3/4 complex to internal nucleosomes at *HML* and *HMR* is thought to

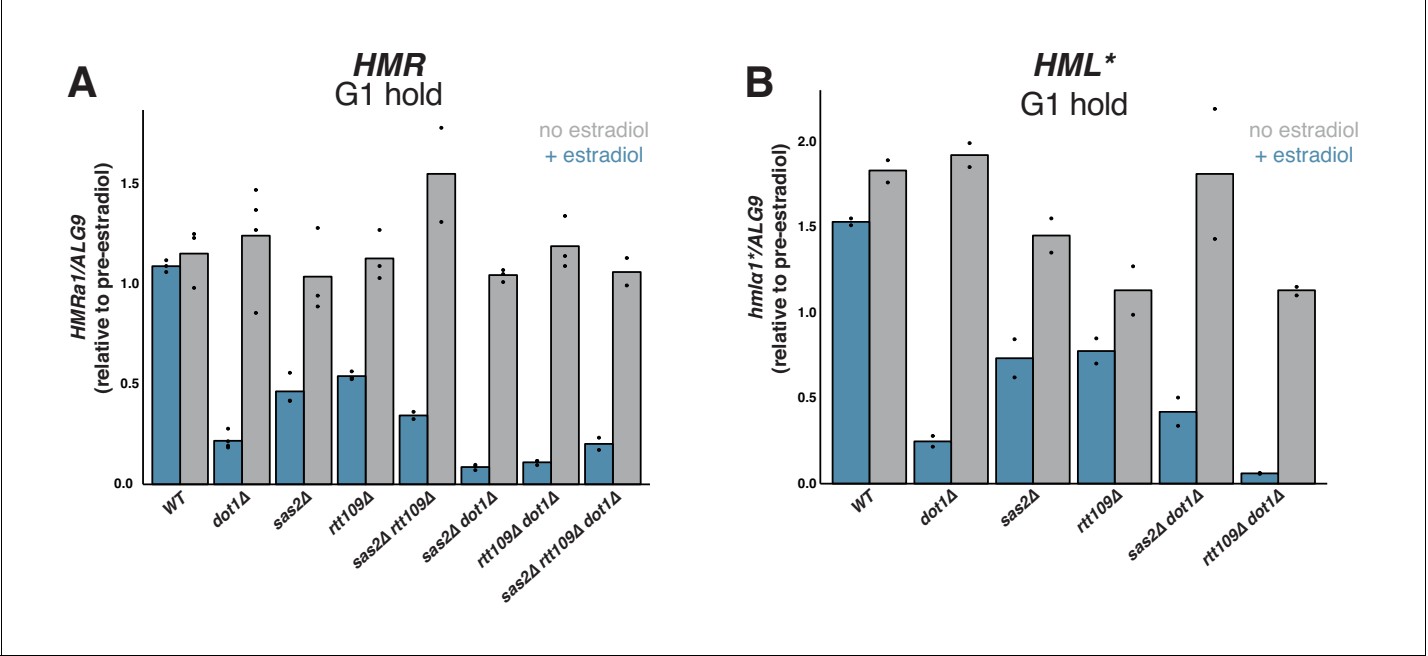

**Figure 5.** Effects of *SAS2* and *RTT109* on silencing establishment in G1. For all strains, cultures were arrested in G1 with α factor, then split, with one sub-culture receiving estradiol and the other receiving ethanol. Silencing was assayed by RT-qPCR 6 hr after additions. Each sample was normalized to its own pre-estradiol value. The following strains were used. WT: JRY12169; *dot1Δ*: JRY12443, JRY12445; *sas2Δ*: JRY12615, JRY12616; *rtt109Δ*: JRY12689, JRY12690; *sas2Δ rtt109Δ*: JRY12765, JRY12766; *sas2Δ dot1Δ*: JRY12618, JRY12619; *rtt109Δ dot1Δ*: JRY12691, JRY12692; *sas2Δ rtt109Δ dot1Δ*: JRY12767, JRY12768. (A) Silencing establishment of *HMRa1* by RT-qPCR. The level of repression observed in each mutant was significantly greater than in wild type (Two-tailed T-test; p<0.005 for each pair-wise comparison). The level of repression observed in *sas2Δ dot1Δ* and *rtt109Δ dot1Δ* double mutants was significantly greater than in the *dot1Δ* single mutant (p<0.01 for each pair-wise comparison), but there was no significant difference between the values from the *dot1Δ* single mutant and the triple mutant *sas2Δ rtt109Δ dot1Δ* (p=0.70). (B) Silencing establishment of *hmlα1** by RT-qPCR in a subset of mutant strains. The level of repression for each mutant was significantly greater than in wild type (p<0.05 for each pair-wise comparison). The level of repression observed in the *rtt109Δ dot1Δ* double mutant was significantly greater than in the *dot1Δ* single mutant (p=0.028), but there was no significant difference between the values from *sas2Δ dot1Δ* double mutant and the *dot1Δ* single mutant (p=0.19).

The online version of this article includes the following source data and figure supplement(s) for figure 5:

**Source data 1.** P-values for comparisons displayed in *Figure 5*.
**Figure supplement 1.** Representative flow cytometry profiles.

drive gene repression at least partly through sterically preventing other proteins from accessing the underlying DNA (*Loo and Rine, 1994*; *Steakley and Rine, 2015*). Repression may also rely on inhibition of specific steps in transcription downstream of activator binding (*Chen and Widom, 2005*; *Gao and Gross, 2008*; *Johnson et al., 2013*). A puzzling observation is that, qualitatively, the nucleation and spread of Sir2, Sir3, and Sir4 to *HML* and *HMR* appears to be cell-cycle-independent, even though the silencing activity of these proteins is clearly dependent on cell cycle progression (*Kirchmaier and Rine, 2006*).

Surprisingly, we found that silencing establishment led to increased Sir4 binding both at silencers and across the silent loci, beginning from a low-level distributed binding that was present even in the absence of Sir3. The weak Sir4 binding across *HML* and *HMR* in *sir3Δ* cells suggested that the full Sir2/3/4 complex was not necessary for the distributed binding of Sir4. Rather, Sir3 appeared to stabilize or otherwise enhance Sir4-nucleosome interactions. Further high-resolution ChIP-seq studies will be needed to determine whether Sir2, Sir3, and Sir4, which localize indistinguishably in wild-type cells (*Thurtle and Rine, 2014*), behave similarly in the absence of the full complex, and during silencing establishment.

Upon induction of *SIR3-EBD*, G1-arrested wild-type cells could robustly recruit Sir4 to *HML* and *HMR* without causing any gene repression at these loci. Passage through S phase led to increased

Sir binding and partial silencing establishment. However, in G1-arrested *dot1Δ* cells, in which Sir4 binding patterns were indistinguishable from G1-arrested wild-type cells, induction of *SIR3-EBD* caused partial silencing establishment. Together, these observations indicate that a key regulated step in building heterochromatin occurred after the major silencing factors were already present at the locus. Two interpretations were compatible with our data. First, the non-repressive Sir4 binding observed in G1 and the repressive Sir4 binding observed in G2/M could differ in some parameter that is not apparent in crosslinking ChIP experiments, such as differences in the on and off rates for Sir4 binding to nucleosomes. Second, Sir binding could be unable to drive transcriptional changes until competing euchromatic marks on chromatin are relieved. Consistent with the latter interpretation, we found that Sir protein binding could lead to changes in H3K79 trimethylation only after S phase. In addition, a prior study of telomeric silencing found that while Sir protein binding was detectable at both repressed and de-repressed telomeres, euchromatic marks, including H3K79me, were found only at de-repressed telomeres, and that, in vitro, H3K79me could disturb silencing without changing Sir protein binding (*Kitada et al., 2012*).

Our smRNA-FISH results showed that silencing establishment proceeds via the gradual tune-down of transcription in individual cells, and that this tune-down occurs over multiple cell cycles. Interestingly, the fraction of cells with zero transcripts after a single S phase (~30%, see *Figure 2B*), is similar to the fraction of cells that established phenotypic silencing after a single division in a previous study (*Osborne et al., 2009*). While these results are not directly comparable, as the previous study assayed silencing at *HML* and used a different induction strategy, one possibility is that phenotypic silencing only arises when transcript number falls to zero in a given cell. This result conflicts with a prior study of silencing establishment in single cells using a fluorescent reporter at *HML*. That study concluded that silencing establishment proceeded via discrete transitions from the 'ON' to the 'OFF' state (*Xu et al., 2006*). However, because that study relied on qualitative assessment of fluorescence intensity in individual cells, it may not have been possible to ascertain intermediate states. Indeed, our data illustrate an inherent limitation of qualitative measurements of single-cell parameters: in the smRNA-FISH images in *Figure 2*, a striking feature is the dichotomy between cells with no transcripts and those with some transcripts. That observation might lead to the conclusion that silencing establishment is caused by the complete shutdown of transcription stochastically in some cells. However, as illustrated by *Figure 2A*, that dichotomy is expected from both an 'all-or-nothing' model and a 'gradual transition' model. It is only through the quantitative analysis that we could see the gradual decrease in transcription in individual cells.

Whether silencing acts through steric occlusion or through a more specific inhibition of some component necessary for transcription, it is difficult to explain how any intermediate in the assembly of a static heterochromatin structure could drive partial repression. The simplest explanation for how partially repressive chromatin could form would be that silencing machinery and transcriptional machinery both come on and off the chromatin, and the establishment of silencing involves a change in the relative rates of those two processes. In that case, histone modifications could be crucial in shifting the balance.

## Euchromatic histone mark removal was a key cell-cycle-regulated step in silencing establishment

Removal of Dot1-deposited methylation of H3K79 was a critical step in silencing establishment. This finding was consistent with earlier studies of cycling cells, which found that *dot1Δ* cells established silencing more quickly than wild-type cells (*Katan-Khaykovich and Struhl, 2005*; *Osborne et al., 2009*). Indeed, Katan-Khaykovich and Struhl proposed a model for silencing establishment in which Sir protein binding and histone deacetylation occur rapidly, followed by slow removal of methylation over several cell cycles, which is consistent with our findings. Removal of H3K79 methylation appears to be the primary reason why cells need to progress through S phase to establish silencing. Dot1 is thought to reduce the Sir3 BAH domain's affinity for the nucleosome core by methylating H3K79 (*Martino et al., 2009*; *Ng et al., 2002a*; *Onishi et al., 2007*). In addition to Dot1 and Sir3 both binding the nucleosome core at H3K79, they also compete for binding to the H4 tail, and deacetylation of the tail by Sir2 is thought to favor Sir3 binding at the expense of Dot1 (*Altaf et al., 2007*). Thus, through modifications at H4K16 and H3K79, transcription and silencing mutually antagonize each other. In a cell in G1, even if Sir2/3/4 are able to displace Dot1 by deacetylating H4K16, H3K79me

will remain until histones are turned over, which seems to explain the S-phase requirement for silencing establishment.

Silencing establishment at *HMR* does not require replication of the locus, as shown by the ability of excised episomes bearing *HMR* but no replication origin to establish silencing in an S-phase-dependent manner (*Kirchmaier and Rine, 2001*; *Li et al., 2001*). This finding presented a major mystery: what S-phase-specific process other than replication fork passage drives silencing establishment? Our results suggest that an influx of H3 molecules lacking methylation at K79 could be the solution. Replication-independent histone exchange can occur throughout the cell cycle (*Dion et al., 2007*; *Rufiange et al., 2007*; *Schlissel and Rine, 2019*), which means that a replicating or non-replicating copy of *HMR* can incorporate histone molecules from the nuclear pool. Outside of S phase, ~90% of all H3 in the nucleus is methylated at K79 (*van Leeuwen et al., 2002*), so histone exchange would likely lead to incorporation of the silencing-refractory methylated form. However, during S phase, a large quantity of newly-synthesized non-methylated H3 is present. Therefore, histone incorporation during S phase through either replication-coupled chromatin assembly or replication-independent histone turnover would lead to incorporation of many H3 molecules that are not methylated at K79. This might explain why silencing establishment can occur at *HMR*, whether it is replicated or not, and why that establishment depends on S phase.

We found that *H3K79R* mutants, which mimicked the non-methylated state of H3K79, also established silencing in G1-arrested cells and did so even more strongly than *dot1Δ* mutants. A simple explanation for this difference in impact of the two mutations could be that Sir3 binds more strongly to arginine than lysine at position 79. Alternatively, Dot1 has been shown to have several methyltransferase-independent functions, and it was possible that one of these functions acted to promote silencing. In particular, Dot1 has recently been shown to possess histone chaperone activity that is independent of its ability to methylate histones (*Lee et al., 2018*). In addition, Dot1 has the methyltransferase-independent ability to stimulate ubiquitination of histone H2B (*van Welsem et al., 2018*). The latter result is particularly interesting, because H2B ubiquitination is itself required for both H3K79 methylation (*Briggs et al., 2002*; *Ng et al., 2002b*) and H3K4 methylation (*Dover et al., 2002*; *Sun and Allis, 2002*). Conflicting reports have pointed to a role of H3K4 methylation in silencing (*Fingerman et al., 2005*; *Mueller et al., 2006*; *Santos-Rosa et al., 2004*). Thus, it is possible that in a *dot1Δ* mutant, the removal of H3K79me per se promotes silencing, but an indirect effect through H2Bub and/or H3K4me partially counteracts the H3K79me effect.

The histone acetyltransferases Sas2 and Rtt109 also had roles in limiting silencing establishment to S phase. Individually, *sas2Δ* and *rtt109Δ* mutations led to partial silencing establishment in G1-arrested cells, and each of these effects was additive with a *dot1Δ* mutation. Acetylation of H4K16 by Sas2, like methylation of H3K79 by Dot1, is critical in distinguishing euchromatin and heterochromatin. Interestingly, in a previous study, while *dot1Δ* sped silencing establishment at *HML*, *sas2Δ* delayed silencing establishment by that assay (*Osborne et al., 2009*). The single-cell α-factor response assay used in that study required cells to fully repress *HML* to gain the **a** mating type identity, whereas our assay used more direct measures of changes in transcription at *HML* and *HMR*. Thus, one explanation consistent with both results is that *sas2Δ* cells begin silencing more readily than wild-type cells, but take more cell cycles to reach full repression. This could be the result of the competing effects of the *sas2Δ* mutation: hypoacetylation of histones at *HML* and *HMR* might increase Sir protein recruitment, while the global pool of hypoacetylated histones can also titrate Sir proteins away from *HML* and *HMR*.

The ability of *rtt109Δ* cells to drive partial silencing establishment in G1-arrested cells was surprising. Like Sas2, Rtt109 binds to Asf1 and acetylates newly-synthesized histones (*Driscoll et al., 2007*), but H3K56 acetylation is removed after S phase by the sirtuins Hst3 and Hst4 (*Celic et al., 2006*). The residual H3K56ac present outside of S phase is due to transcription-coupled histone turnover, which incorporates new histones marked with H3K56ac (*Rufiange et al., 2007*). A negative role for H3K56 acetylation in silencing has been observed, although this has not been well-characterized (*Dodson and Rine, 2015*; *Miller et al., 2008*). One simple model is that H3K56ac favors transcription, and thus impedes silencing establishment. However, given genome-wide acetylation and deacetylation of H3K56, indirect effects cannot be excluded.

## Silencing establishment occurred via similar mechanisms at different loci

The mechanism of repression at the two silent mating type loci, *HML* and *HMR*, is generally assumed to be quite similar, but there are mutations that cause effects only at one of the two loci, and others that cause divergent phenotypes between the two loci (*Park and Szostak, 1990*; *Ehrenhofer-Murray et al., 1997*; Yan and Rine, unpublished). Earlier studies concluded that cell-cycle requirements for silencing establishment differed at *HML* and *HMR* (*Ren et al., 2010*; *Lazarus and Holmes, 2011*) In contrast, in both wild-type cells and the mutant conditions we tested, both loci behaved similarly. A major innovation that distinguished our studies from the prior studies was our use of a mutant *HML* that allowed unambiguous study of its expression by ensuring that α1 and α2 proteins would not be made. This strategy removed the strong repressive effect that the **a**1/α2 repressor has on transcription from the *HML* promoter, which was a confounding influence in earlier experimental designs that could have led to apparent cell-cycle independent silencing of *HML*. In the course of this work, we found that the *HML* promoter is subject to hyperactivation by α factor, which further complicates studies of silencing establishment at the locus. More work is clearly needed to fully understand how silencing establishment is regulated at *HML*.

We did note one distinction between *HML\** *and* HMR. After a single S phase, H3K79 trimethylation was depleted from *HMR* and *hmlα1\**, but not from *hmlα2\**. Given that H3K79 methylation is a major regulator of silencing establishment, this could explain the observation that *hmlα2\** was also the gene whose silencing was weakest after a single S phase of silencing establishment. The interrelation between promoter strength, transcription-coupled histone modification, and silencing remains a fascinating topic for future study.

This fundamental similarity between *HML* and *HMR* in silencing establishment was further evidenced by the lack of an effect of the tRNA gene adjacent to *HMR,* or the tRNA gene's binding partner, cohesin, loss of either of which were reported to allow early silencing establishment in previous studies (*Lau et al., 2002*; *Lazarus and Holmes, 2011*). The reason behind the differences between our results and those of the previous studies was not clear. We did observe subtle silencing-independent fluctuations in *HMRa1* expression through the cell cycle, which may have confounded earlier results that relied on non-quantitative RT-PCR assays (data not shown). We cannot exclude the possibility that differences between *SIR3-EBD* and earlier inducible alleles contributed to the different results, as temperature, metabolism, and hormone addition could each affect silencing or the cell cycle in unappreciated ways. Lazarus and Holmes's use of the galactose promoter to drive *SIR3* expression would alter Sir3 concentration and the stoichiometry of the SIR complex, both of which would be expected to be important parameters in regulating silencing establishment.

## Do the contributions of *DOT1*, *SAS2*, and *RTT109* completely resolve the cell-cycle requirement for silencing establishment?

In *dot1Δ* mutants, S phase still dramatically accelerated silencing establishment, indicating that some feature of S phase beyond H3K79me removal was important in those cells. In addition, we found no case in which silencing establishment in G1-arrested cells matched the degree of silencing observed after overnight growth in estradiol. However, the ~90% repression observed in, for example G1-arrested *H3K79R* cells should be sufficient to completely turn off transcription at *HML* and *HMR* in the majority of cells (see *Figure 2*). The quantitative gap in the level of silencing seen at steady state and that which is achieved in the experiments reported here could reflect a requirement for further cell-cycle steps or more time to complete silencing establishment. Others have identified a cell-cycle window between G2/M and G1 that contributes to silencing establishment (*Lau et al., 2002*), and none of our data were inconsistent with that result. Identifying mutant conditions in which G1-arrested cells and cycling cells establish silencing at an equal rate will be required before the cell-cycle-regulated establishment of silencing is fully understood. In addition, future studies should address whether the antagonistic effects of euchromatic histone modifications on silencing establishment can be counteracted by increasing SIR complex concentration.

Together, our data suggest that silencing establishment cannot proceed without removal of histone modifications that favor transcription. In this view, at any stage of the cell cycle, Sir proteins can bind to *HML* and *HMR*. Passage through S phase leads to incorporation of new

histones, which, crucially, lack H3K79 methylation. This decrease of H3K79me by half leads to both further Sir binding and decreased transcription. However, one cell cycle is not sufficient to fully deplete activating marks, and successive passages through S phase complete the process of silencing establishment.

# Materials and methods

## Key resources table

| Reagent type (species) or resource | Designation | Source or reference | Identifiers | Additional information |
|---|---|---|---|---|
| Strain, strain background (*Saccharomyces cerevisiae*) | Various | This paper | NCBITaxon:4932 | See *Supplementary file 1a* |
| Antibody | Anti-c-myc beads (mouse monoclonal) | Thermo Fisher Scientific | Cat # 88842 | 50 µL per IP |
| Antibody | Anti-H3K79me3 (rabbit polyclonal) | Diagenode | Cat # C15410068 | 5 µL per IP |
| Antibody | Anti-ERα (rabbit polyclonal) | Santa Cruz Biotechnology | Cat # sc-8002, RRID:AB_627558 | 25 µL per IP |
| Antibody | Anti-Hexokinase (Rabbit polyclonal) | Rockland | Cat # 100–4159, RRID:AB_219918 | (1:20,000) |
| Antibody | Anti-V5 (Mouse monoclonal) | Thermo Fisher Scientific | Cat # R960-25, RRID:AB_2556564 | (1:2,500) |
| Antibody | IRDye 800CW anti-mouse (Goat polyclonal) | Li-Cor | Cat # 926–32210, RRID:AB_621842 | (1:20,000) |
| Antibody | IRDye 680RD anti-rabbit (Goat polyclonal) | Li-Cor | Cat # 926–68070, RRID:AB_10956588 | (1:20,000) |
| Recombinant DNA reagent | HML* | This paper | | Mutated allele of HML |
| Recombinant DNA reagent | SIR3-EBD | This paper | | Fusion protein of Sir3 and EBD of mammalian ERα |
| Sequence-based reagent | Various oligonucleotides | This paper | qPCR primers | See *Supplementary file 1b* |
| Sequence-based reagent | smRNA-FISH probes | Biosearch Technologies | | See *Supplementary file 1c* |
| Commercial assay or kit | RNEasy Mini Kit | Qiagen | Cat # 74104 | |
| Commercial assay or kit | Qiaquick PCR purification kit | Qiagen | Cat # 28104 | |
| Commercial assay or kit | NEBNext Ultra II Library prep kit | NEB | Cat # 37645L | |
| Commercial assay or kit | Superscript III reverse transcriptase kit | Thermo Fisher Scientific | Cat # 18080044 | |
| Commercial assay or kit | DyNamo HS SYBR Green qPCR kit | Thermo Fisher Scientific | Cat # F410L | |
| Peptide, recombinant protein | Catalase | Sigma-Aldrich | Cat # C3515 | |
| Peptide, recombinant protein | Proteinase K | NEB | Cat # P8107S | |
| Peptide, recombinant protein | Glucose oxidase | Sigma-Aldrich | Cat # G2133 | |
| Peptide, recombinant protein | Zymolyase-100T | VWR | Cat # IC320932 | |

*Continued on next page*

*Continued*

| Reagent type (species) or resource | Designation | Source or reference | Identifiers | Additional information |
|---|---|---|---|---|
| Peptide, recombinant protein | Protein A beads | Thermo Fisher Scientific | Cat # 10002D | |
| Peptide, recombinant protein | α-factor peptide | Elim Bio | | |
| Peptide, recombinant protein | Uracil-DNA Glycosylase | Thermo Fisher Scientific | Cat # EN0362 | |
| Peptide, recombinant protein | Micrococcal nuclease | Worthington Biochemical | Cat # LS004798 | |
| Commercial assay or kit | RNase-Free DNase Set | Qiagen | Cat # 79254 | |
| Chemical compound, drug | Nocodazole | Sigma-Aldrich | Cat # M1404 | |
| Chemical compound, drug | β-estradiol | Sigma-Aldrich | Cat # E8875 | |
| Chemical compound, drug | cOmplete EDTA-free protease inhibitor | Simga-Aldrich | Cat # 11873580001 | |
| Chemical compound, drug | 3-indoleacetic acid | Sigma-Aldrich | Cat # I2886 | |
| Software, algorithm | SAMtools | doi:10.1093/bioinformatics/btp352 | RRID:SCR_002105 | |
| Software, algorithm | Ggplot2 | doi:10.1007/978-0-387-98141-3 | RRID:SCR_014601 | |
| Software, algorithm | Bowtie2 | doi:10.1038/nmeth.1923 | RRID:SCR_005476 | |
| Software, algorithm | FISH-quant | doi:10.1038/nmeth.2406 | | |
| Software, algorithm | FlowJo | BD Life Sciences | RRID:SCR_008520 | |

## Yeast strains

Strains used in this study are listed in *Supplementary file 1a*. All strains were derived from the W303 background using standard genetic techniques (*Dunham et al., 2015*; *Gietz and Schiestl, 2007*). Deletions were generated using one-step replacement with marker cassettes (*Goldstein and McCusker, 1999*; *Gueldener et al., 2002*). The *sir3-8* allele was introduced by a cross to the strain Y3451 (*Xu et al., 2006*). The tRNA gene *tT(AGU)C* was seamlessly deleted using the 'delitto per-fetto' technique as described previously (*Storici and Resnick, 2006*). The *MCD1-AID* strain was gen-erated by first inserting *O.s.TIR1* at the *HIS3* locus by transforming cells with PmeI-digested pTIR2 (*Eng et al., 2014*). Then, *3xV5-AID2:KanMX* was amplified from pVG497 (a gift from Vincent Guacci and Douglas Koshland) with primers that included homology to *MCD1*, followed by transformation. The mutant allele *HML\** was synthesized as a DNA gene block (Integrated DNA Technologies) and integrated using CRISPR-Cas9 technology as previously described (*Brothers and Rine, 2019*). The *EBD* sequence was amplified by PCR from *cre-EBD78* in the strain UCC5181 (*Lindstrom and Gottschling, 2009*) with primers that included homology to *SIR3*, then transformed using CRISPR-Cas9. Mutations of *HHT1* and *HHT2* were generated using CRISPR-Cas9, with oligonucleotide repair templates. For all mutant analyses, at least two independent transformants or meiotic segregants were tested.

## Culture growth and cell-cycle manipulations

All experiments were performed on cells growing in yeast extract peptone + 2% dextrose (YPD) at 37°C, which led to more switch-like behavior for *SIR3-EBD* than growth at 30°C. For biological repli-cates, independent cultures were started from the same strain or from two isogenic strains. For steady-state measurements, cells were grown overnight in YPD, then diluted and grown in fresh YPD to a density of ~2–8×10$^6$ cells/mL. For cell-cycle control experiments, cells were grown overnight in YPD, followed by dilution and growth in fresh YPD for ≥3 hr until cultures reached a density of ~2×10$^6$ cells/mL. Then, α factor (synthesized by Elim Biopharmaceuticals; Hayward, CA) was added to a final concentration of 10 nM and the cultures were incubated for ~2 hr until >90% of cells

appeared unbudded. For *rtt109Δ* strains, this incubation was ~3 hr. For experiments with prolonged α-factor arrests, additional α factor was added every ~2 hr to maintain the arrest. To release cells from α-factor arrest, protease from *Streptomyces griseus* (Sigma-Aldrich P5147; St. Louis, MO) was added to the media at a final concentration of 0.1 mg/mL. To re-arrest cells at G2/M, nocodazole (Sigma-Aldrich M1404) was added to a final concentration of 15 μg/mL. For *SIR3-EBD* induction, β-estradiol (Sigma-Aldrich E8875) was added to a final concentration of 50 μM from a 10 mM stock in ethanol. For Mcd1-AID depletion, 3-indoleacetic acid (auxin; Sigma-Aldrich I2886) was added to a final concentration of 750 μM from a 1 M DMSO stock. For *sir3-8* temperature shifts, cells were grown continuously at 37°C, then shifted to 24°C for the length of the experiment.

## RNA extraction and RT-qPCR

For each sample, at least ~$1 \times 10^7$ cells were collected by centrifugation and RNA was purified using an RNeasy Mini Kit (Qiagen 74104; Hilden, Germany), including on-column DNase digestion (Cat No. 79254), according to manufacturer's recommendations. 2 μg of RNA was reverse transcribed using SuperScript III reverse transcriptase (Thermo Fisher Scientific 18080044; Waltham, MA) and an 'anchored' oligo-dT primer (an equimolar mixture of primers with the sequence $T_{20}VN$, where V represents any non-T base). qPCR was performed using the DyNAmo HS SYBR Green qPCR kit (Thermo Fisher Scientific F410L), including a Uracil-DNA Glycosylase (Thermo Fisher Scientific EN0362) treatment, and samples were run using an Agilent Mx3000P thermocycler. Oligonucleotides used for qPCR are listed in *Supplementary file 1b*. cDNA abundance was calculated using a standard curve and normalized to the reference gene *ALG9*. Each reaction was performed in triplicate, and a matched non-reverse-transcribed sample was included for each sample.

## Chromatin immunoprecipitation and sequencing

For MNase ChIP experiments, ~$5 \times 10^8$ cells were crosslinked in 1% formaldehyde at room temperature for 60 min (*Figure 3A and B*) or 15 min (all other figures). Following a 5 min quench in 300 mM glycine, cells were washed twice in ice-cold TBS and twice in ice-cold FA lysis buffer (50 mM HEPES, pH 7.5; 150 mM NaCl, 1 mM EDTA, 1% Triton, 0.1% sodium deoxycholate) + 0.1% SDS + protease inhibitors (cOmplete EDTA-free protease inhibitor cocktail, Sigma-Aldrich 11873580001). Cell pellets were then either flash frozen or lysed. For lysis, cell pellets were resuspended in 1 mL FA lysis buffer without EDTA + 0.1% SDS and ~500 μL 0.5 mm zirconia/Silica beads (BioSpec Products; Bartlesville, OK) were added. Cells were lysed using a FastPrep-24 5G (MP Biomedicals; Irvine, CA) with 6.0 m/s beating for 20 s followed by 2 min on ice, repeated four times total. Lysate was transferred to a new microcentrifuge tube, then spun at 4°C for 15 min at 17,000 rcf. The pellet was resuspended in 1 mL FA lysis buffer without EDTA + 0.1% SDS + 2 mM $CaCl_2$. The samples were incubated at 37°C for 5 min, followed by addition of 4 U MNase (Worthington Biochemical LS004798; Lakewood, NJ) per $1 \times 10^7$ cells. Samples were placed on an end-over-end rotator at 37°C for 20 min, followed by addition of 1.25 mM EDTA to quench the digestion. Samples were spun at 4°C for 10 min at 17,000 rcf, and the supernatant containing fragmented chromatin was saved.

For sonication ChIP experiments (*Figure 3—figure supplement 2* and *Figure 4—figure supplement 2* only), fragmentation was performed as above, except following lysis and spin-down, the pellet was resuspended in ~500 μL FA Lysis Buffer + 0.1% SDS, split between two tubes and sonicated using a Bioruptor Pico (Diagenode Inc; Denville, NJ) for 15 cycles of 30 s ON followed by 30 s OFF. Following sonication, samples were spun at 4°C for 10 min at 17,000 RCF and supernatants were re-pooled.

The fragmented chromatin was split, saving 50 μL as the input sample, and using the remaining ~900 μL for immunoprecipitation. IP for Sir4-13xMyc was performed using 50 μL Pierce Anti-c-myc magnetic beads (Thermo Fisher Scientific 88843) per sample. Beads were equilibrated by washing 5x in FA Lysis buffer + 0.1% SDS + 0.05% Tween, then resuspended in in 50 μL per sample of FA Lysis buffer + 0.1% SDS + 0.05% Tween. For Sir3-EBD, IP was performed using 25 μL of rabbit polyclonal anti-ERα (sc-8002; Santa Cruz Biotechnology, Inc; Dallas, TX). For H3K79me3, IP was performed using 5 μL of rabbit polyclonal anti-H3K79me3 (C15410068; Diagenode). All immunoprecipitations were performed in an end-over-end rotator at 4°C overnight in the presence of 0.5 mg/mL BSA (NEB B9000S; Ipswich, MA). For the Sir3-EBD and H3K79me3 IPs, samples were incubated for 1 hr at 4°C with 50 μL Dynabeads Protein A magnetic beads (Thermo Fisher Scientific 10002D),

equilibrated as described above for Anti-c-myc beads. The following washes were performed, placing each sample on an end-over-end rotator for ~5 min between each: 2x washes with FA Lysis + 0.1% SDS + 0.05% Tween; 2x washes with Wash Buffer #1 (FA Lysis buffer + 0.25 M NaCl + 0.1% SDS + 0.05% Tween); 2x washes with Wash Buffer #2 (10 mM Tris, pH 8; 0.25 M LiCl; 0.5% NP-40; 0.5% sodium deoxycholate; 1 mM EDTA + 0.1% SDS + 0.05% Tween); and 1x wash with TE + 0.05% Tween. Samples were eluted by adding 100 µL TE + 1% SDS to the beads and incubating at 65°C for 10 min with gentle shaking. The eluate was saved, and the elution was repeated for a total eluate volume of 200 µL. Input samples were brought to a total volume of 200 µL with TE + 1% SDS. IP and input samples were incubated with 10 µL of 800 U/mL Proteinase K (NEB P8107S) at 37°C for 2 hr, followed by overnight incubation at 65°C to reverse crosslinking.

DNA was purified using a QIAquick PCR purification kit (Qiagen 28104). Libraries were prepared for high-throughput sequencing using the Ultra II DNA Library Prep kit (NEB E7645L). For IP samples, the entire purified sample was used in the library prep reaction. For the input samples, 10 ng were used. Samples were multiplexed and paired-end sequencing was performed using either a MiniSeq or NovaSeq 6000 (Illumina; San Diego, CA).

Sequencing reads were aligned using Bowtie2 (*Langmead and Salzberg, 2012*) to a reference genome derived from SacCer3, modified to include the mutant *HML\** and *matΔ*. Reads were normalized to the non-heterochromatic genome-wide median (i.e., to the genome-wide median excluding rDNA, subtelomeric regions, and all of chromosome III). For the MNase ChIP-seq experiments, only mononucleosome-sized fragments (130–180 bp) were mapped. Analysis was performed using custom Python scripts (*Source code 1*, *Source code 2*) and displayed using IGV (*Thorvaldsdóttir et al., 2013*). For coverage calculations in *Figure 4—figure supplement 2*, read coverage for each gene was calculated using the bedcov function of SAMtools (*Li et al., 2009*), and then normalized to the length of each gene and mean coverage for all genes. Scatter plots were generated using ggplot2 (*Wickham, 2016*). All raw and processed sequencing data were deposited at the GEO under the accession number GSE150737.

## Single-molecule RNA fluorescence in situ hybridization

For smRNA-FISH experiments, cells were grown and cell-cycle arrests were performed as described above. Preparation of samples for imaging was as previously described (*Chen et al., 2018*). The only modification from that protocol was the concentration of zymolyase used: for cycling cells, 5 µL of 1.25 mg/mL zymolyase-100T (VWR IC320932; Radnor, PA) was used during the spheroplasting; for arrested cells, 5 µL of 2.5 mg/mL zymolyase-100T was used. Probes were synthesized by Stellaris (Biosearch Technologies; Novato, CA). Probes for *cre* and *KAP104* were previously described (*Dodson and Rine, 2015*). The sequences of all probes, including newly-designed probes for *a1* are listed in *Supplementary file 1c*. Probes for *cre* and *a1* were coupled to Quasar 670 dye, while probes for *KAP104* were coupled to CAL Fluor Red 590 dye. Probes for *cre* and *KAP104* were used at a final concentration of 100 nM. Probes for *a1* were used at a final concentration of 25 nM.

Imaging was performed on an Axio Observer Z1 inverted microscope (Zeiss; Oberkochen, Germany) equipped with a Plan-Apochromat 63x oil-immersion objective (Zeiss), pE-300 ultra illumination system (CoolLED; Andover, UK), MS-2000 XYZ automated stage (Applied Scientific Instrumentation; Eugene, OR), and 95B sCMOS camera (Teledyne Photometrics; Tucson, AZ). The following filter sets were used: for Quasar 670, Cy5 Narrow Excitation (Chroma Cat No. 49009; Bellows Falls, VT); for CAL Fluor Red 590, filter set 43 HE (Zeiss); for DAPI, multiband 405/488/594 filter set (Semrock Part No. LF405/488/594-A-000; Rochester, NY). The microscope was controlled using Micro-manager software (*Edelstein et al., 2014*). Z-stack images were taken with a total height of 10 µm and a step size of 0.2 µm. Manual cell outlining and automatic spot detection was performed using FISH-quant (*Mueller et al., 2013*) and data were plotted using ggplot2 (*Wickham, 2016*). Representative images were generated using FIJI (*Schindelin et al., 2012*).

## Protein extraction and immunoblotting

Protein extraction and immunoblotting were performed as previously described (*Brothers and Rine, 2019*). The primary antibodies used were 1:20,000 Rabbit anti-Hexokinase (Rockland #100–4159; Limerick, PA) and 1:2,500 Mouse anti-V5 (Thermo Fisher Scientific R960-25). The secondary

antibodies used were 1:20,000 IRDye 800CW Goat anti-Mouse (Li-Cor; Lincoln, NE) and 1:20,000 IRDye 680RD Goat anti-Rabbit (Li-Cor).

### Flow cytometry

For every cell-cycle experiment, an aliquot of cells was taken for flow-cytometry analysis of DNA content to monitor cell-cycle stage. Sample preparation and flow cytometry was performed as described previously (*Schlissel and Rine, 2019*). A representative sample of flow cytometry profiles is shown in *Figure 5—figure supplement 1*.

## Acknowledgements

We are grateful to all members of the Rine laboratory and Koshland laboratory for helpful discussions and comments on this work. We thank Vincent Guacci for providing plasmids and for his valuable guidance, along with Lorenzo Costantino, on performing cell-cycle manipulations. We thank Itamar Patek for his help with strain construction. We give special thanks to Marc Fouet for his generous assistance with microscopy. This work relied on the Vincent J Coates Genomics Sequencing Laboratory at UC Berkeley, supported by NIH S10 OD018174 Instrumentation Grant. This work was funded by grants from the National Institutes of Health to J.R. (GM31105, GM120374). DG received support from NIH training grants (T32 GM007127, T32 HG000047) and a National Science Foundation Graduate Research Fellowship (Grant No. 1752814).

## Additional information

### Funding

| Funder | Grant reference number | Author |
|---|---|---|
| National Institutes of Health | R01 GM31105 | Jasper Rine |
| National Institutes of Health | R01 GM120374 | Jasper Rine |
| National Institutes of Health | T32 GM007127 | Davis Goodnight |
| National Institutes of Health | T32 HG000047 | Davis Goodnight |
| National Science Foundation | DGE 1752814 | Davis Goodnight |

The funders had no role in study design, data collection and interpretation, or the decision to submit the work for publication.

### Author contributions

Davis Goodnight, Conceptualization, Resources, Data curation, Software, Formal analysis, Validation, Investigation, Visualization, Methodology, Writing - original draft, Project administration, Writing - review and editing; Jasper Rine, Conceptualization, Resources, Supervision, Funding acquisition, Project administration, Writing - review and editing

### Author ORCIDs

Davis Goodnight ⓘD https://orcid.org/0000-0002-5423-1424
Jasper Rine ⓘD https://orcid.org/0000-0003-2297-9814

### Decision letter and Author response

Decision letter https://doi.org/10.7554/eLife.58910.sa1
Author response https://doi.org/10.7554/eLife.58910.sa2

## Additional files

### Supplementary files

- Source code 1. Python script for bedgraph generation.
- Source code 2. Python script for bedgraph normalization.

- Supplementary file 1. Supplementary File 1a: Yeast strains used in this study. All strains listed were generated for this study and derived from the W303 background. Supplementary File 1b: Oligonucleotides used for RT-qPCR Supplementary File 1c: Probes used for smRNA-FISH.
- Transparent reporting form

### Data availability

Sequencing data have been deposited in the GEO under the accession number GSE150737.

The following dataset was generated:

| Author(s) | Year | Dataset title | Dataset URL | Database and Identifier |
|---|---|---|---|---|
| Goodnight D, Rine J | 2020 | S-phase independent silencing establishment in Saccharomyces cerevisiae | https://www.ncbi.nlm. nih.gov/geo/query/acc. cgi?acc=GSE150737 | NCBI Gene Expression Omnibus, GSE150737 |

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
