## [Decision Letter]

**Acceptance summary:**

This manuscript addresses a long-standing issue regarding the role of S-phase passage in establishing transcriptional silencing without the need for replication itself. The support provided for a model in which existing chromatin carrying histones with pro-transcription modifications must be turned over by passive mechanisms fed by the unmodified histones available only during S phase provides an elegant solution to this persistent question, and your use of orthogonal, robust methods provides high confidence in the conclusions. The results have high relevance to models for epigenetic switches outside the yeast system and will therefore have broad impact.

**Decision letter after peer review:**

Thank you for submitting your article "S-phase-independent silencing establishment in *Saccharomyces cerevisiae*" for consideration by *eLife*. Your article has been reviewed by three peer reviewers, including Tim Formosa as the Reviewing Editor and Reviewer #1, and the evaluation has been overseen by Kevin Struhl as the Senior Editor. The following individuals involved in review of your submission have agreed to reveal their identity: Rohinton T Kamakaka (Reviewer #2); Danesh Moazed (Reviewer #3).

The reviewers have discussed the reviews with one another and the Reviewing Editor has drafted this decision to help you prepare a revised submission.

Summary:

Goodnight and Rine provide new insight into a longstanding puzzle regarding the establishment of transcriptionally silent heterochromatin in yeast. Using higher resolution ChIP and single-molecule RNA detection methods, they show that cells must pass through S phase multiple times to promote establishment of silenced heterochromatin, and suggest that it is the availability of a large pool of histones lacking modifications associated with transcriptional activation during S phase that is the key requirement met during this period but not during other parts of the cell cycle. These new insights have broad impact on understanding how epigenetic states are established and inherited.

The data are of high quality and are clearly presented. However, several issues were raised during review that should be considered by the authors prior to publication.

Essential revisions:

1) The authors should be more careful to present the contributions of other groups to the model they propose more accurately, especially concerning the use of mutations in *HML* here relative to work published by the Holmes and Sternglanz labs. Further, the distinction between Sir protein occupancy and timing of silencing and the requirement of multiple cell divisions for complete silencing that are key results here was also noted by the Struhl lab in their 2005 paper.

2) Reviewers felt that the manuscript would be strengthened by more discussion of their current in vivo findings in the context of what is known about the biochemical activities of Sir proteins in reconstitution experiments in vitro. The results in the following publications seem particularly relevant: Onishi et al., 2007; Johnson et al., 2009; Johnson et al., 2013; Kitada et al., 2012.

3) The Sir3-EBD strategy has certain advantages but may also be the source of some discrepancies with previous work. Ideally, the authors should consider some side-by-side comparisons of the new method with older approaches. However, it would be sufficient here to note the possible complications introduced by using a different approach and to use more restrained language in describing its superiority over past approaches to avoid implying that these results invalidate previous findings.

4) Several points of interest were raised that may be addressable by changes to the text, or the authors may consider them important enough to provide new experimental data; the latter is encouraged but not required. These include the following questions: Is the 60% repression observed in one cell cycle enough to block mating in a phenotypic assay? Can silencing be established in G1 by overexpression of Sir2/3/4? Could the proposal suggested here that differences between *HML* and *HMR* are due to promoter differences be tested by swapping the a/α information at these sites? Does Sir3 occupancy match the Sir4 occupancy reported here?

5) The authors might consider providing some additional context for the non-expert reader. For example, the authors state that "…so that the α1 and α2 proteins were never made, even when *HML* was de-repressed." The logic here may be unclear to those less versed in the mechanics of mating type switching, so it would be useful to provide further explanation.

---

## [Author Response]

Essential revisions:1) The authors should be more careful to present the contributions of other groups to the model they propose more accurately, especially concerning the use of mutations in HML here relative to work published by the Holmes and Sternglanz labs. Further, the distinction between Sir protein occupancy and timing of silencing and the requirement of multiple cell divisions for complete silencing that are key results here was also noted by the Struhl lab in their 2005 paper.

First of all, we thank the reviewers for catching what was our unintentional failure to emphasize the work of others that were part of the foundation and motivation of this study. We agree that the conclusions reached by our study agree with and build on the results of Katan-Khaykovich and Struhl and have introduced further context in the fourth paragraph of the Introduction and in the first paragraph of the subsection “Euchromatic histone mark removal was a key cell-cycle-regulated step in silencing establishment”, clarifying what is new in this study and what was identified by them in 2005.

With respect to the work of the Holmes and Sternglanz labs, we tried to take a nuanced approach as we believe that several key experiments from those labs were uninterpretable because of their lack of consideration of the issue of the a1/α2 repressor. In our effort to not offer offense to the authors of those studies, we erred on the side of understatement. Studying silencing establishment at *HML* inherently difficult due to the confounding effects from the a1/α2 repressor that we believe led the Sternglanz/Holmes experiments astray. In our revision we have been more explicit about their conclusions and have emphasized the limitations of ours caused by the Ste12-driven hyperactivation of *HML.* We have introduced a comment in the Discussion that emphasizes how our *HML* data have limitations (subsection “Silencing establishment occurred via similar mechanisms at different loci”, first paragraph.

2) Reviewers felt that the manuscript would be strengthened by more discussion of their current in vivo findings in the context of what is known about the biochemical activities of Sir proteins in reconstitution experiments in vitro. The results in the following publications seem particularly relevant: Onishi et al., 2007; Johnson et al., 2009; Johnson et al., 2013; Kitada et al., 2012.

We thank the reviewers for pointing out this relevant literature. We agree that many of the results presented here are buttressed by in vitro data about silencing, particularly with respect to the impact of H3K79 methylation on Sir3 binding. We have increased our citations of the suggested studies (subsections “Silencing establishment occurred by tuning down transcription in individual cells after Sir proteins were bound” and “Euchromatic histone mark removal was a key cell-cycle-regulated step in silencing establishment).

3) The Sir3-EBD strategy has certain advantages but may also be the source of some discrepancies with previous work. Ideally, the authors should consider some side-by-side comparisons of the new method with older approaches. However, it would be sufficient here to note the possible complications introduced by using a different approach and to use more restrained language in describing its superiority over past approaches to avoid implying that these results invalidate previous findings.

We agree with the reviewers about the importance of offering some validation that use of Sir3-EBD given that we are making comparisons to earlier work of others with *sir3-8*. We have tested the comparability directly and have now have included a new experiment in Figure 4E demonstrating that *dot1∆* can lead to S-phase-independent silencing establishment using the classic *sir3-8* allele with data that are essentially indistinguishable with the data from Sir3-EBD. We thank the reviewers for encouraging us to address this as other readers would likely have had the same reservations. Although we have now removed this unknown, there may still be unknown unknowns, and in our Discussion, we emphasized the potential role that different alleles of Sir3 could play in different experiments (see subsection “Silencing establishment occurred via similar mechanisms at different loci”, last paragraph).

4) Several points of interest were raised that may be addressable by changes to the text, or the authors may consider them important enough to provide new experimental data; the latter is encouraged but not required. These include the following questions: Is the 60% repression observed in one cell cycle enough to block mating in a phenotypic assay?

This specific experiment (assaying mating) is not possible in our strains, since we rely on α-factor responsiveness of cells with or without intact silencing to synchronize cells. However, a previous study by Osborne et al. speaks to the comparison between RNA measurements and mating ability. We have added a more explicit discussion of this study (subsection “Silencing establishment occurred by tuning down transcription in individual cells after Sir proteins were bound”, fourth paragraph), which we feel addresses the point: while direct comparisons between the two studies are difficult, we believe mating ability is probably restored only when ~0 transcripts per cell are present.

Can silencing be established in G1 by overexpression of Sir2/3/4?

We have long been interested in this question, but so far have not been able to achieve simultaneous overexpression of all three proteins in a balanced way. Moreover, work by others emphasize the potential importance of stoichiometry over overproduction. For example, while mild increases in Sir4 dosage can speed silencing establishment, strong overexpression leads to weakened silencing (Larin et al., PMID: 26587833). We are working to develop an overexpression strategy that we are confident maintains proper complex stoichiometry, but we have no idea when that will be successful. We have included a comment in the Discussion (subsection “Do the contributions of DOT1, SAS2, and RTT109 completely resolve the cell-cycle requirement for silencing establishment?”) that raises this as a topic for future study.

Could the proposal suggested here that differences between HML and HMR are due to promoter differences be tested by swapping the a/α information at these sites?

We made no claims about the source of differences seen at *HML* vs. *HMR*. We agree that strength of transcription is likely to be key, but our basic finding is that the differences in silencing establishment at *HML* and *HMR* are minor: *HMRa1* is repressed ~0.6x after a single S-phase, while *hmlα2** is repressed ~0.4x. We do not view this difference as a fruitful avenue for future study. However, we added a new experiment in Figure 4—figure supplement 2, which aimed to answer a different question (namely, is H3K79 methylation depletion in fact limited only to S phase), but revealed a difference between *HML* and *HMR*: *hmlα2** was far less able to deplete H3K79me after 1 S phase than was *HMRa1*. This could be due to a difference in strength of transcription (which drives H3K79 methylation). We have included this comment on possible contribution of promoter strength in our discussion of the new result (subsection “Silencing establishment occurred via similar mechanisms at different loci”, second paragraph).

Does Sir3 occupancy match the Sir4 occupancy reported here?

We have introduced Figure 3—figure supplement 3, which shows that Sir3-EBD binds indistinguishably from Sir4-13xMyc at steady state. In addition, we have resolved an issue that some find puzzling about the lack of a Sir protein signal at the silencers in the MNase seq data. We have added a new experiment showing the Sir4 occupancy determined by sonication ChIP-seq (Figure 3—figure supplement 2). This experiment revealed that, as we suspected, our MNase ChIP-seq data severely under-report Sir4 binding to the silencers of *HMR* and *HML* due to their MNase sensitivity. We have adjusted our claims in the text to reflect this new information, but these data do not change any of our conclusions.

5) The authors might consider providing some additional context for the non-expert reader. For example, the authors state that "…so that the α1 and α2 proteins were never made, even when HML was de-repressed." The logic here may be unclear to those less versed in the mechanics of mating type switching, so it would be useful to provide further explanation.

We agree that this paragraph would be difficult for a non-expert and have added further context in the fourth paragraph of the subsection “S phase as a critical window for silencing establishment”.